# Gender differences in the effect of self-rated health (SRH) on all-cause mortality and specific causes of mortality among individuals aged 50 years and older

Insun Ryou[1], Yujin Cho[2], Hyung-Jin Yoon[3], Minseon Park[2]*

1 Department of Family Medicine, Ewha Womans University Medical Center, Ewha Womans University School of Medicine, Seoul, Republic of Korea, 2 Department of Family Medicine, Seoul National University Hospital, Seoul National University College of Medicine, Seoul, Republic of Korea, 3 Department of Biomedical Engineering, Seoul National University College of Medicine, Seoul, Republic of Korea

* msp20476@hanmail.net

**Data Availability Statement:** All relevant data are within the paper.

**Funding:** The authors received no specific funding for this work.

## Abstract

Although different gender associations between self-rated health (SRH) and mortality have been reported, the results of the respective studies have been inconsistent and little is known about the cause-specific relation of mortality with SRH by gender. Therefore, to evaluate the gender differences in all-cause or specific causes of mortality by SRH, this retrospective cohort study was conducted using the data of 19,770 Korean adults aged 50 years and over who underwent health screening at Seoul National University Hospital between March 1995 and December 2008. SRH was surveyed using a simple questionnaire, and the all-cause mortality and cause-specific mortality were followed up from baseline screening until December 31, 2016. Results showed that the relationship between SRH and all-cause mortality differed by gender, and the differences also varied depending on the cause of death. In men, the adjusted hazard ratio (aHR) of all-cause mortality was higher in the poor SRH group than the very good SRH groups even after adjustment for socio-demographic, clinical, and behavioral risk factors (aHR:1.97, 95% CI 1.51–2.56), and these results were similar to those for cancer, cardiovascular, and respiratory disease mortalities (aHR:1.52, 95% CI 0.93–2.50; aHR: 2.11, 95% CI 1.19–3.74; aHR:10.30, 95% CI 2.39–44.44, respectively). However, in women, the association between SRH and all-cause mortality was insignificant, and inverse relationships were found for cardiovascular and respiratory disease mortalities in the poor and very good SRH groups. Cancer mortality had a positive relation with SRH (aHR: 1.14, 95% CI 0.75–1.72; aHR: 2.58, 95% CI 1.03–6.48; aHR: 0.49, 95% CI 0.24–0.98; aHR: 0.15, 95% CI 0.04–0.57: all-cause, cancer, cardiovascular, and respiratory disease mortalities, respectively). Clinicians need to take these gender differences by SRH into account when evaluating the health status of over-middle aged adults.

**Competing interests:** The authors have declared that no competing interests exist.

## Introduction

Self-rated health (SRH) has been reported as a predictor of mortality [1–3] even after controlling for related confounding factors [4–8]. In addition, several studies have suggested that SRH also predicts specific causes of mortality, such as cancer, respiratory, and cardiovascular diseases. Most studies showed relatively consistent results that poor SRH was associated with increased risk of all-cause mortality and cause-specific mortality, mainly in older adults [2, 6, 8–12]. Population-based studies showed that poor SRH was related to an increased risk of mortality from cancer and respiratory diseases [9, 10, 13]. One population-based prospective cohort study in the UK showed that SRH was a strong predictor of cardiovascular deaths after adjusting for socio-demographic, clinical, and behavioral risk factors [14].

However, there have been a variety of inconsistent results regarding the relevance of SRH to mortality, such as the magnitude of the effect of the relationship and the differences in results according to confounding factors like age, sex, and sociodemographic and clinical data. Among these, gender differences showed the most notable inconsistency [15]. Some studies showed a strong association of SRH with mortality only in males [16], but others suggested that the association of SRH and mortality was not affected by gender [6, 15].

In assessing their general health status, men usually tend to reflect serious and life-threatening diseases, but women tend to reflect both life-threatening and non-life-threatening health status. Therefore, some researchers suggested that different processes of assessing general health state entailed different relationships between SRH and health outcomes according to gender [17–19]. Moreover, it was unknown which specific cause of mortality brought about this gender difference.

Therefore, this study was conducted to evaluate gender differences in the association between SRH and mortality, and more particularly to identify the specific cause of mortality in relation to differences in gender associations among healthy middle-aged Korean populations.

## Materials and methods

### Study population

We retrospectively collected data for individuals who had received medical check-ups and had completed the SRH questionnaire at the Health Screening Center of Seoul National University Hospital between May 1995 and December 2008. Of the 50,690 people who received health screenings during the period, 44,537 persons whose survival and death data confirmed by December 2016 and who completed SRH questionnaires were extracted. Among them, 39,380 were included after excluding those with missing socio-demographic, clinical, or behavioral data, ($n = 4,647$) and those who died within one year after the medical check-ups ($n = 56$). We also excluded 20,055 individuals who were under 50 years old. Therefore, 19,770 individuals (9,944 men and 9,826 women) in total were included in the final analysis.

### Assessment of SRH

SRH was evaluated by completing a questionnaire with the following question when conducting a health checkup at the Seoul National University Hospital Medical Center: "In general, how do you think your health is?" The responses were categorized into four levels: very good, good, fair, and poor.

### Ascertainment of covariates

The baseline survey involved a 30-item questionnaire assessing health status, health-related behavior, past medical history, and socio-demographic information. Socio-demographic

variables included age, sex, education level, income level, occupation classification, and marital status. Education level was categorized as follows: "elementary school graduate," "middle school graduate," "high school graduate," and "college degree." Income level was categorized into quintiles. Occupational status was classified as "no occupation," "white collar," and "blue collar." Marital status was categorized as "single," "married," "divorced/separated," and "widowed." Health-related behavior variables included smoking, regular drinking, nighttime sleep duration, and exercise. Smoking status was categorized into three groups: "never," "ex-smoker," and "current smokers." A regular drinker was defined as someone who drinks alcoholic beverages at least once a week. Regular exercisers were defined as those who exercised more than 20 minutes at a time at least three times a week, which was estimated from the questions in the 30-item questionnaire about the kind of regular physical exercise and the frequency and duration of each physical activity per week during the month before the examination. Clinical variables included body mass index (BMI), diagnosis of hypertension or diabetes, prognostic nutritional index (PNI), maximum $O_2$ uptake (VO2max), and the Brief Encounter Psychological Instrument-Korean version (BEPSI-K) score. VO2max was measured by a graded exercise test with bicycle ergometer to assess the individual's fitness level. Height and weight were measured after overnight fasting in light clothing, and the body mass index (BMI) was calculated as (weight (kg) / height (m)$^2$). Blood pressure was measured using an automated blood pressure device after each individual had been seated for at least 20 minutes. At the baseline screening, we obtained 12hr overnight fasting blood samples. Hypertension was defined as systolic blood pressure $\geq$ 140mmHg or diastolic blood pressure $\geq$ 90mmHg at baseline examination, previous history of hypertension, or current administration of anti-hypertensive medications. Diabetes was defined as plasma glucose $\geq$ 126mg/dL at the time of examination, previous history of diabetes, or current administration of anti-diabetic medications. The PNI (prognostic nutritional index) was calculated as a combination of the albumin and total lymphocyte counts and scored as 0 ($\geq$ 45) and 1 ($<$ 45). We used BEPSI-K to assess the severity of stress. BEPSI-K is a self-reported questionnaire with five questions whose scores are averaged for the final score. Individuals were categorized into three groups by their scores: Low (score $<$ 1.8), moderate (1.8 $\leq$ score $<$ 2.8), and high (score $\geq$ 2.8). We used data on SRH obtained at the baseline screening (1995–2008).

## Ascertainment of mortality

Mortality for the present study was evaluated by following individuals from the baseline screening until death in 2016. All deceased individuals were ascertained through the records of the national death certificate files in Korea. Because of the possibility of death of a patient who had stage 0 cancer, asymptomatic heart disease, or un-diagnosed unknown disease at screening day, we excluded deaths within one year after screening to eliminate these causes.

The cause of death was classified into four categories: "cancer," "cardiovascular diseases (CVD)," "respiratory diseases," and "others" using the International Classification of Diseases (ICD) 10[th] revision. Cancer death was defined using codes C00–C97, cardiovascular deaths using codes I00–I99, and respiratory deaths using codes J00–J99. Other causes included trauma, genitourinary, gastrointestinal, musculoskeletal, congenital diseases, and dementia, besides infection.

## Statistical analysis

All statistical analyses were performed using STATA version 14.0. Quantitative data are given as mean ± standard deviation (*SD*). The chi-square test was used to compare categorical variables, whereas one-way ANOVA was used for continuous variables. The predictive value of

SRH for all-cause mortality and specific causes of mortality was estimated using Cox proportional hazard models with 95% confidence intervals (CIs). All causes of deaths were adjusted for age, body mass index (BMI), smoking, drinking, exercise, diagnosis of hypertension or diabetes, marital status, education level, occupational status, PNI, VO2max, BEPSI-K sleep time, history of cancer, total cholesterol level, fasting blood glucose level, and GFR. Significance was set at $p$ (two-sided) $< 0.05$.

## Results

Because of the differences in the perception of SRH in men and women, we analyzed most of our results separately by gender. At the baseline screening, 9,944 subjects were men and 9,826 were women. Their baseline characteristics are shown in Table 1.

According to Table 1, women reported worse SRH than men. Among men, 712 (7.4%) assessed their SRH as "very good," 2,935 (29.5%) as "good," 5,242 (52.7%) as "fair," and 1,050 (10.5%) as "poor," while among women, 390 (4.0%) assessed their SRH as "very good," 1,652 (16.8%) as "good," 5,348 (54.4%) as "fair," and 2,436 (24.8%) as "poor." Male respondents with very good SRH tended to be older and more educated; have more income; be more likely to hold white collar jobs; be non-current smokers (never, ex-smoker), regular drinkers, and regular exercisers; be less frequently diagnosed with diabetes; have lower fasting blood glucose (FBG) levels, higher total cholesterol levels, and lower GFR; be more obese; and have higher VO2max, lower stress levels, and 7 to 8 hours of nighttime sleep duration. Women with very good SRH tended to be older and more educated; have higher income; be more likely to hold white collar jobs (though most of them had no occupation); not be regular drinkers; be regular exercisers; be less frequently diagnosed with hypertension and diabetes; have lower FBG levels, higher total cholesterol levels, and lower GFR; be more obese; and have higher VO2max, lower stress levels, and 7 to 8 hours of nighttime sleep duration.

Table 2 shows the crude incidence rates of all-cause mortality and specific causes of mortality. During a median follow-up of 15.4 years, 2,263 of the 19,770 individuals (19.2%) had died. The most common cause of death was cancer (44.5%). Among CVD, ischemic heart disease was the most common ($n = 114$, 5.0% of deaths), followed by hemorrhage stroke ($n = 80$, 3.5% of deaths) and ischemic stroke ($n = 61$, 2.7% of deaths). The incidence rate of all-cause mortality according to SRH was the lowest for very good SRH and the highest for poor SRH (782 vs 910 vs 965 vs 1,866 for very good, good, fair, and poor SRH, respectively) in males. These results were similar for cancer, cardiovascular disease, and respiratory diseases in males. However, the incidence rate of all-cause mortality was higher in those with very good SRH than with good or fair SRH (441 vs 360 vs 425 vs 667 for very good, good, fair, and poor SRH, respectively) in females. These results were similar for cardiovascular and respiratory diseases but not for cancer, which was lowest for very good SRH and the highest for poor SRH (82 vs 188 vs 184 vs 256 for very good, good, fair, and poor SRH, respectively) in females.

Table 3 shows the crude hazard ratio (HR) of all-cause mortality according to baseline characteristics by gender.

There were no gender differences in the other factors besides SRH. In men, the effects of SRH on all-cause mortality gradually increased from very good to poor (HR 1.16 vs 1.23 vs 2.36; good, fair, and poor SRH, respectively), whereas in women, the HR of good and fair SRH was lower than that of very good SRH, although the risk of all-cause mortality was the highest for poor SRH (HR 0.84 vs 0.99 vs 1.46; good, fair, and poor SRH, respectively).

The subjects with poor SRH, old age ($\geq 65$ years), current smokers, having history of hypertension, diabetes, cancer, having more than 126mg/dL of FBG level, lower GFR, longer sleep time, poor nutrition, and high stress level had a higher HR than their counterparts. However,

**Table 1. Baseline characteristics by categories of SRH according to gender.**

| | Male (N = 9,944) | | | | | Female (N = 9,826) | | | | |
|---|---|---|---|---|---|---|---|---|---|---|
| | Very good (n = 712) | Good (n = 2,935) | Fair (n = 5,247) | Poor (n = 1,050) | P value | Very good (n = 390) | Good (n = 1,652) | Fair (n = 5,348) | Poor (n = 2,436) | P value |
| **Age at screening, years, (SD)** | 59.7 (6.7) | 58.9 (6.2) | 58.1 (5.8) | 58.0 (5.7) | < 0.001 | 58.1 (6.2) | 57.7 (5.7) | 57.5 (16.9) | 57.1 (5.3) | < 0.001 |
| **Smoking, n, (%)** | | | | | | | | | | |
| Never | 208 (29.2) | 689 (23.5) | 1,137 (21.7) | 193 (18.4) | < 0.001 | 372 (95.4) | 1,569 (95.0) | 5,127 (95.9) | 2,295 (94.2) | 0.028 |
| Ex-smoker | 313 (44.0) | 1,307 (44.5) | 2,155 (41.1) | 388 (37.0) | | 10 (2.6) | 33 (2.0) | 89 (1.7) | 49 (2.0) | |
| Current | 191 (26.8) | 939 (32.0) | 1,955 (37.3) | 469 (44.7) | | 8 (2.1) | 50 (3.0) | 132 (2.5) | 92 (3.8) | |
| **Drinking, n, (%)** | | | | | | | | | | |
| No drinking | 219 (30.8) | 861 (29.3) | 1,738 (33.1) | 514 (49.0) | < 0.001 | 311 (79.7) | 1,391 (84.2) | 4,631 (86.6) | 2,183 (89.6) | < 0.001 |
| Regular drinker[a] | 493 (69.2) | 2,074 (70.7) | 3,509 (66.9) | 536 (51.1) | | 79 (20.3) | 261 (15.8) | 717 (13.4) | 253 (10.4) | |
| **Exercise[b], n, (%)** | | | | | | | | | | |
| No | 234 (33.7) | 1,057 (37.0) | 2,164 (41.9) | 532 (51.7) | < 0.001 | 139 (36.1) | 611 (37.5) | 2,101 (39.8) | 1,252 (52.1) | < 0.001 |
| Yes | 460 (66.3) | 1,800 (63.0) | 3,000 (58.1) | 497 (48.3) | | 246 (63.9) | 1,019 (62.5) | 3,184 (60.3) | 1,151 (47.9) | |
| **Diabetes[c], n, (%)** | | | | | | | | | | |
| No | 636 (89.3) | 2,618 (89.2) | 4,636 (88.4) | 858 (81.7) | < 0.001 | 374 (95.9) | 1,560 (94.4) | 5,045 (94.3) | 2,148 (88.2) | < 0.001 |
| Yes | 76 (10.7) | 317 (10.8) | 611 (11.6) | 192 (18.3) | | 16 (4.1) | 92 (5.6) | 303 (5.7) | 288 (11.8) | |
| **Hypertension[d], n, (%)** | | | | | | | | | | |
| No | 594 (83.4) | 2,462 (83.9) | 4,352 (82.9) | 869 (82.8) | 0.707 | 336 (86.2) | 1,419 (85.9) | 4,467 (83.5) | 1,922 (78.9) | < 0.001 |
| Yes | 118 (16.6) | 473 (16.1) | 895 (17.1) | 181 (17.2) | | 54 (13.9) | 233 (14.1) | 881 (16.5) | 514 (21.1) | |
| **History of cancer, n, (%)** | | | | | | | | | | |
| No | 685 (96.3) | 2,801 (95.6) | 4,994 (95.4) | 1,007 (96.2) | 0.532 | 378 (97.2) | 1,608 (97.6) | 5,194 (97.2) | 2,351 (96.8) | 0.412 |
| Yes | 26 (3.7) | 129 (4.4) | 240 (4.6) | 40 (3.8) | | 11 (2.8) | 39 (2.4) | 149 (2.8) | 79 (3.3) | |
| **Systolic BP[e], mmHg, (SD)** | 136.5 (20.0) | 135.8 (20.3) | 134.5 (20.2) | 130.5 (21.0) | 0.464 | 136.1 (20.1) | 136.1 (21.2) | 135.8 (21.7) | 135.9 (22.2) | 0.035 |
| **Diastolic BP, mmHg, (SD)** | 82.7 (11.9) | 82.5 (12.2) | 81.8 (12.10) | 79.5 (12.5) | 0.435 | 80.7 (12.3) | 80.5 (11.8) | 80.6 (11.9) | 80.9 (12.2) | 0.384 |
| **Fasting blood glucose, mg/dL, (SD)** | 102.9 (24.9) | 102.9 (27.8) | 105.1 (31.0) | 106.8 (36.7) | < 0.001 | 96.9 (20.1) | 97.4 (22.6) | 98.4 (23.0) | 101.9 (30.6) | < 0.001 |
| **Serum total cholesterol, mg/dL, (SD)** | 205.5 (35.0) | 203.3 (36.7) | 201.5 (35.3) | 195.8 (38.9) | < 0.001 | 217.9 (38.9) | 217.1 (38.4) | 215.3 (38.6) | 212.8 (41.4) | < 0.001 |
| **eGFR using CKD-EPI, ml/min per 1.73 m², (SD)** | 78.1 (12.2) | 79.2 (12.3) | 80.5 (13.0) | 81.3 (14.1) | < 0.001 | 80.4 (13.4) | 81.3 (13.5) | 81.7 (13.1) | 82.8 (14.0) | 0.006 |
| **BMI[f], n, (%)** | | | | | | | | | | |
| Underweight | 2 (0.3) | 27 (0.9) | 102 (1.9) | 89 (8.5) | < 0.001 | 3 (0.8) | 11 (0.7) | 71 (1.3) | 76 (3.1) | < 0.001 |
| Normal | 150 (21.1) | 708 (24.1) | 1,772 (33.8) | 455 (43.3) | | 101 (25.9) | 523 (31.7) | 1,760 (32.9) | 791 (32.5) | |
| Overweight | 223 (31.3) | 983 (33.5) | 1,591 (30.3) | 259 (24.7) | | 122 (31.3) | 492 (29.8) | 1,512 (28.3) | 623 (25.6) | |
| Obese | 337 (47.3) | 1,217 (41.5) | 1,782 (34.0) | 247 (23.5) | | 164 (42.1) | 626 (37.9) | 2,005 (37.5) | 946 (38.8) | |
| **Nighttime sleep duration, n, (%)** | | | | | | | | | | |

*(Continued)*

**Table 1.** (Continued)

| | Male (N = 9,944) | | | | | Female (N = 9,826) | | | | |
|---|---|---|---|---|---|---|---|---|---|---|
| | Very good (n = 712) | Good (n = 2,935) | Fair (n = 5,247) | Poor (n = 1,050) | P value | Very good (n = 390) | Good (n = 1,652) | Fair (n = 5,348) | Poor (n = 2,436) | P value |
| < 6h | 286 (40.2) | 1,081 (36.8) | 1,953 (37.2) | 443 (42.2) | < 0.001 | 154 (39.5) | 655 (39.7) | 2,318 (43.3) | 1,224 (50.3) | < 0.001 |
| 7–8h | 399 (56.0) | 1,755 (59.8) | 3,136 (59.8) | 549 (52.3) | | 216 (55.4) | 930 (56.3) | 2,899 (54.2) | 1,084 (44.5) | |
| ≥ 9h | 27 (3.8) | 99 (3.4) | 158 (3.0) | 58 (5.52) | | 20 (5.1) | 67 (4.1) | 131 (2.5) | 128 (5.25) | |
| **PNI[g], n, (%)** | | | | | | | | | | |
| 0 | 245 (34.4) | 1,008 (34.3) | 1,811 (34.5) | 340 (32.4) | 0.613 | 116 (29.7) | 493 (29.8) | 1,628 (30.4) | 738 (30.3) | 0.966 |
| 1 | 467 (65.6) | 1,927 (65.7) | 3,436 (65.5) | 710 (67.6) | | 274 (70.3) | 1,159 (70.2) | 3,720 (69.6) | 1,698 (69.7) | |
| **VO2max[h], n, (%)** | | | | | | | | | | |
| Low | 201 (28.2) | 742 (25.28) | 1,432 (27.3) | 352 (33.5) | < 0.001 | 82 (21.0) | 370 (22.4) | 1,415 (26.5) | 940 (38.6) | < 0.001 |
| Moderate | 233 (32.7) | 1,109 (37.79) | 1,972 (37.6) | 338 (32.2) | | 143 (36.7) | 588 (35.6) | 1,758 (32.9) | 676 (27.8) | |
| High | 278 (39.0) | 1,084 (36.93) | 1,843 (35.1) | 360 (34.3) | | 165 (42.30) | 694 (42) | 2,175 (40.7) | 820 (33.7) | |
| **BEPSI-K[i], n, (%)** | | | | | | | | | | |
| Low | 545 (76.5) | 2,195 (74.8) | 3,516 (67.0) | 552 (52.6) | < 0.001 | 270 (69.2) | 1,132 (68.5) | 3,098 (57.9) | 1,070 (43.9) | < 0.001 |
| Moderate | 109 (15.3) | 523 (17.8) | 1,227 (23.4) | 324 (30.9) | | 86 (22.1) | 351 (21.3) | 1,509 (28.2) | 825 (33.9) | |
| High | 58 (8.2) | 217 (7.4) | 504 (9.6) | 174 (16.6) | | 34 (8.7) | 169 (10.2) | 741 (13.9) | 541 (22.2) | |
| **Education level, n, (%)** | | | | | | | | | | |
| Elementary school graduate | 64 (9.0) | 316 (10.8) | 801 (15.3) | 293 (27.9) | < 0.001 | 75 (19.2) | 416 (25.2) | 1,817(34.0) | 1,297 (53.2) | < 0.001 |
| Middle school graduate | 87 (12.2) | 324 (11.0) | 794 (15.1) | 198 (18.9) | | 73 (18.7) | 282 (17.1) | 1,055 (19.7) | 436 (17.9) | |
| High school graduate | 177 (24.9) | 821 (28.0) | 1,580 (30.1) | 311 (29.6) | | 121 (31.0) | 498 (30.2) | 1,525 (28.5) | 489 (20.1) | |
| College degree | 384 (53.9) | 1,474 (50.2) | 2,072 (39.5) | 248 (23.6) | | 121 (31.0) | 456 (27.6) | 951 (17.8) | 214 (8.8) | |
| **Income level, n, (%)** | | | | | | | | | | |
| 1st Quartile | 59 (8.3) | 263 (9.0) | 666 (12.7) | 248 (23.6) | < 0.001 | 68 (17.4) | 234 (14.2) | 1,029 (19.2) | 747 (30.7) | < 0.001 |
| 2nd Quartile | 138 (19.4) | 663 (22.6) | 1,489 (28.4) | 341 (32.5) | | 85 (21.8) | 423 (25.6) | 1,764 (33.0) | 848 (34.8) | |
| 3rd Quartile | 204 (28.7) | 1,021 (34.8) | 1,732 (33.0) | 286 (27.2) | | 105 (26.9) | 530 (32.1) | 1,547 (28.9) | 542 (22.3) | |
| 4th Quartile | 311 (43.70) | 988 (33.7) | 1,360 (25.9) | 175 (16.7) | | 132 (33.9) | 465 (28.2) | 1,008 (18.9) | 299 (12.3) | |
| **Occupation classification, n, (%)** | | | | | | | | | | |
| No occupation | 133 (18.7) | 591 (20.1) | 1,058 (20.2) | 255 (24.3) | < 0.001 | 242 (62.1) | 1,173 (71.0) | 4,063 (76.0) | 1,880 (77.2) | < 0.001 |
| White collar | 387 (54.4) | 1,404 (47.8) | 2,147 (40.9) | 291 (27.7) | | 80 (20.5) | 205 (12.4) | 310 (5.8) | 102 (4.2) | |
| Blue collar | 192 (27.0) | 940 (32.0) | 2,042 (38.9) | 504 (48.0) | | 68 (17.4) | 274 (16.6) | 975 (18.2) | 454 (18.6) | |
| **Marital status, n, (%)** | | | | | | | | | | |

*(Continued)*

**Table 1.** (Continued)

| | Male (N = 9,944) | | | | | Female (N = 9,826) | | | | |
|---|---|---|---|---|---|---|---|---|---|---|
| | Very good (n = 712) | Good (n = 2,935) | Fair (n = 5,247) | Poor (n = 1,050) | P value | Very good (n = 390) | Good (n = 1,652) | Fair (n = 5,348) | Poor (n = 2,436) | P value |
| Single | 6 (0.8) | 17 (0.6) | 23 (0.4) | 6 (0.6) | < 0.001 | 13 (3.3) | 32 (1.9) | 45 (0.8) | 16 (0.7) | < 0.001 |
| Married | 666 (93.5) | 2,788 (95.0) | 5,060 (96.4) | 999 (95.1) | | 263 (67.4) | 1,275 (77.2) | 4,227 (79.0) | 1,895 (77.8) | |
| Divorced/separated | 23 (3.2) | 68 (2.3) | 57 (1.1) | 22 (2.1) | | 46 (11.8) | 85 (5.2) | 187 (3.5) | 92 (3.8) | |
| Widowed | 17 (2.4) | 62 (2.1) | 107 (2.0) | 23 (2.2) | | 68 (17.4) | 260 (15.7) | 889 (16.6) | 433 (17.8) | |

Age is shown as mean value ± standard deviation (SD).

[a] Regular drinker: person drinking alcoholic beverages at least once a week.

[b] Exercise: exercise at least three times a week and more than 20 minutes at one time.

[c] Diabetes: plasma glucose ≥ 126mg/dL, previous history of diabetes.

[d] Hypertension: systolic blood pressure ≥ 140mmHg or diastolic blood pressure ≥ 90mmHg at examination, previous history of hypertension, or current administration of antihypertensive (anti-HTN) medications.

[e] BP: Blood pressure.

[f] BMI: Body mass index.

[g] Prognostic Nutritional Index (PNI): calculated as 10 x serum albumin (g/dL) + 0.005 x total lymphocyte count (/mL), scored as 0(≥ 45) or 1(< 45).

[h] VO2max: maximum O2 uptake was measured by graded exercise test with bicycle ergometer, and it was categorized as low (VO2max ≤ 21mL/kg/min for men / VO2max ≤ 10mL/kg/min for women), moderate (21mL/kg/min ≤ VO2max ≤ 27mL/kg/min for men / 10mL/kg/min ≤ VO2max ≤ 18mL/kg/min for women), high (VO2max ≥ 28mL/kg/min for men / VO2max ≥ 19mL/kg/min for women).

[i] BEPSI-K(Brief Encounter Psychosocial Instrument, Korean version) categorized as low (BEPSI-K < 1.8), moderate (1.8 ≤ BEPSI-K < 2.8), high (BEPSI-K ≥ 2.8)

regular drinkers, regular exercisers, more obese, more educated, higher income, white collar, married, and higher VO2max had lower HRs than the opposite.

Table 4 shows the HR of all-cause mortality and specific cause of mortality by gender. In Model 1, age, BMI, smoking, drinking, and socio-demographical factors were adjusted. In

**Table 2. Incidence rate of all-cause and specific causes of mortality according to SRH by gender.**

| | Total | Male | | | | | Female | | | | |
|---|---|---|---|---|---|---|---|---|---|---|---|
| | All-cause mortality | All-cause mortality | Specific cause of mortality | | | | All-cause mortality | Specific cause of mortality | | | |
| | | | Cancer | Cardiovascular disease | Respiratory disease | Others | | Cancer | Cardiovascular disease | Respiratory disease | Others |
| **Very good** | | | | | | | | | | | |
| Event | 111 | 84 | 44 | 17 | 2 | 21 | 27 | 5 | 11 | 4 | 7 |
| IR[a] | **658** | **782** | **410** | **158** | **19** | **195** | **441** | **82** | **180** | **65** | **114** |
| **Good** | | | | | | | | | | | |
| Event | 492 | 400 | 195 | 62 | 24 | 119 | 92 | 48 | 19 | 5 | 20 |
| IR | **707** | **910** | **443** | **141** | **55** | **271** | **360** | **188** | **74** | **20** | **78** |
| **Fair** | | | | | | | | | | | |
| Event | 1,111 | 760 | 350 | 140 | 58 | 212 | 351 | 152 | 76 | 13 | 110 |
| IR | **688** | **965** | **444** | **178** | **74** | **269** | **425** | **184** | **92** | **16** | **133** |
| **Poor** | | | | | | | | | | | |
| Event | 549 | 288 | 114 | 64 | 39 | 71 | 261 | 100 | 51 | 8 | 102 |
| IR | **1006** | **1866** | **739** | **415** | **253** | **460** | **667** | **256** | **130** | **20** | **261** |

[a]IR: Incidence rate (event/100,000 person year)

**Table 3. Hazard ratio of all-cause mortality by categories of baseline characteristics.**

| | Male (N = 9,944) | | Female (N = 9,826) | |
|---|---|---|---|---|
| | n (% or SD) | HR (95% CI) | n (% or SD) | HR (95% CI) |
| **Self-rated health, n (%)** | | | | |
| Very good | 712 (7.2) | 1 | 390 (4.0) | 1 |
| Good | 2,935 (29.5) | 1.16 (0.92–1.47) | 1,652 (16.8) | 0.84 (0.54–1.28) |
| Fair | 5,247 (52.8) | 1.23 (0.98–1.54) | 5,348 (54.4) | 0.99 (0.67–1.46) |
| Poor | 1,050 (10.6) | 2.36 (1.85–3.00) | 2,436 (24.8) | 1.46 (0.98–2.16) |
| **Age at screening, years (SD)** | | | | |
| 50–64 | 8,396 (84.4) | 1 | 8,780 (89.4) | 1 |
| ≥ 65 | 1,548 (15.6) | 3.71 (3.34–4.13) | 1,046 (10.7) | 4.20 (3.58–4.93) |
| **Smoking, n (%)** | | | | |
| Never | 2,227 (22.4) | 1 | 9,363 (95.3) | 1 |
| Ex-smoker | 4,163 (41.9) | 1.20 (1.04–1.39) | 181 (1.8) | 1.89 (1.25–2.86) |
| Current | 3,554 (35.7) | 1.59 (1.38–1.83) | 282 (2.9) | 1.97 (1.43–2.73) |
| **Drinking, n (%)** | | | | |
| No drinking | 3,332 (33.5) | 1 | 8,516 (86.7) | 1 |
| Regular drinker[a] | 6,612 (66.5) | 0.75 (0.68–0.83) | 1,310 (13.3) | 0.78 (0.61–0.99) |
| **Exercise[b], n (%)** | | | | |
| No | 3,987 (40.9) | 1 | 4,103 (42.3) | 1 |
| Yes | 5,757 (59.1) | 0.86 (0.77–0.95) | 5,600 (57.7) | 0.82 (0.71–0.96) |
| **Diabetes[c], n (%)** | | | | |
| No | 8,748 (88.0) | 1 | 9,127 (92.9) | 1 |
| Yes | 1,196 (12.0) | 1.62 (1.42–1.85) | 699 (7.1) | 2.59 (2.13–3.16) |
| **Hypertension[d], n (%)** | | | | |
| No | 8,277 (83.2) | 1 | 8,144 (82.9) | 1 |
| Yes | 1,667 (16.8) | 1.45 (1.28–1.64) | 1,682 (17.1) | 1.66 (1.40–1.98) |
| **History of cancer, n (%)** | | | | |
| No | 9,487 (95.6) | 1 | 9,531 (97.2) | 1 |
| Yes | 435 (4.4) | 2.28 (1.90–2.73) | 278 (2.8) | 3.59 (2.73–4.71) |
| **Systolic BP[e], mmHg, (SD)** | | | | |
| < 140 | 6,157 (61.9) | 1 | 5,884 (59.9) | 1 |
| ≥ 140 | 3,787 (38.1) | 1.42 (1.29–1.58) | 3,942 (40.1) | 1.41 (1.22–1.63) |
| **Diastolic BP, mmHg, (SD)** | | | | |
| < 90 | 7,413 (74.6) | 1 | 7,659 (78.0) | 1 |
| ≥ 90 | 2,531 (25.5) | 0.96 (0.85–1.07) | 2,167 (22.1) | 1.18 (1.00–1.40) |
| **Fasting blood glucose, mg/dL (SD)** | | | | |
| < 100 | 5,704 (57.4) | 1 | 6,601 (67.2) | 1 |
| 100–125 | 3,076 (30.9) | 0.99 (0.88–1.11) | 2,540 (25.9) | 1.17 (0.99–1.39) |
| ≥ 126 | 1,164 (11.7) | 1.58 (1.37–1.81) | 685 (7.0) | 2.27 (1.83–2.82) |
| **Serum total cholesterol, mg/dL, (SD)** | | | | |
| < 200 | 4,861 (48.9) | 1 | 3,470 (35.3) | 1 |
| 200–239 | 3,747 (37.7) | 0.85 (0.76–0.95) | 4,021 (40.9) | 0.84 (0.71–1.00) |
| ≥ 240 | 1,336 (13.4) | 1.00 (0.86–1.16) | 2,335 (23.8) | 1.04 (0.86–1.25) |
| **eGFR using CKD-EPI, ml/min per 1.73 m² (SD)** | | | | |
| ≥ 90 | 2,365 (23.8) | 1 | 2,933 (29.9) | 1 |
| 60–89 | 7,064 (71.0) | 1.01 (0.90–1.13) | 6,474 (65.9) | 1.32 (1.12–1.55) |
| < 60 | 515 (5.2) | 2.75 (2.28–3.31) | 419 (4.3) | 3.00 (2.24–4.02) |
| **BMI[f], n (%)** | | | | |
| Underweight | 220 (2.2) | 1 | 161 (1.6) | 1 |
| Normal | 3,085 (31.0) | 0.47 (0.37–0.60) | 3,175 (32.3) | 0.66 (0.41–1.06) |
| Overweight | 3,056 (30.7) | 0.35 (0.28–0.45) | 2,749 (28.0) | 0.55 (0.34–0.89) |
| Obese | 3,583 (36.0) | 0.32 (0.25–0.40) | 3,741 (38.1) | 0.70 (0.44–1.13) |

*(Continued)*

**Table 3.** (Continued)

| | Male (N = 9,944) | | Female (N = 9,826) | |
|---|---|---|---|---|
| | **n (% or SD)** | **HR (95% CI)** | **n (% or SD)** | **HR (95% CI)** |
| **Nighttime sleep duration, n (%)** | | | | |
| < 6h | 3,763 (37.8) | 1.05 (0.95–1.17) | 4,351 (44.3) | 0.99 (0.86–1.15) |
| 7–8h | 5,839 (58.7) | 1 | 5,129 (52.2) | 1 |
| ≥ 9h | 342 (3.4) | 1.56 (1.24–1.97) | 346 (3.5) | 1.30 (0.93–1.83) |
| **PNI[g], n (%)** | | | | |
| 0 | 3,404 (34.2) | 1 | 2,975 (30.3) | 1 |
| 1 | 6,540 (65.8) | 1.53 (1.36–1.72 | 6,851 (69.7) | 1.11 (0.94–1.30) |
| **VO2max[h], n (%)** | | | | |
| Low | 2,727 (27.4) | 1 | 2,807 (28.6) | 1 |
| Moderate | 3,652 (36.7) | 0.59 (0.52–0.66) | 3,165 (32.2) | 0.58 (0.49–0.69) |
| High | 3,565 (35.9) | 0.50 (0.44–0.57) | 3,854 (39.2) | 0.45 (0.37–0.54) |
| **BEPSI-K[i], n (%)** | | | | |
| Low | 6,808 (68.5) | 1 | 5,570 (56.7) | 1 |
| Moderate | 2,183 (22.0) | 1.06 (0.94–1.19) | 2,771 (28.2) | 1.00 (0.85–1.18) |
| High | 953 (9.6) | 1.11 (0.93–1.33) | 1,485 (15.1) | 1.13 (0.92–1.39) |
| **Education level, n (%)** | | | | |
| Elementary school graduate | 1,474 (14.8) | 1 | 3,605 (36.7) | 1 |
| Middle school graduate | 1,403 (14.1) | 0.66 (0.57–0.78) | 1,846 (18.8) | 0.70 (0.57–0.85) |
| High school graduate | 2,889 (29.1) | 0.54 (0.47–0.62) | 2,633 (26.8) | 0.67 (0.55–0.80) |
| College degree | 4,178 (42.0) | 0.41 (0.36–0.47) | 1,742 (17.7) | 0.47 (0.37–0.60) |
| **Income level, n (%)** | | | | |
| 1st Quartile, % | 1,236 (12.4) | 1 | 2,078 (21.2) | 1 |
| 2nd Quartile, % | 2,631 (26.5) | 0.57 (0.50–0.66) | 3,120 (31.8) | 0.75 (0.63–0.89) |
| 3rd Quartile, % | 3,243 (32.6) | 0.39 (0.34–0.45) | 2,724 (27.7) | 0.53 (0.43–0.65) |
| 4th Quartile, % | 2,834 (28.5) | 0.31 (0.26–0.36) | 1,904 (19.4) | 0.55 (0.43–0.70) |
| **Occupation classification, n (%)** | | | | |
| No occupation | 2,037 (20.5) | 1 | 7,358 (74.9) | 1 |
| White collar | 4,229 (42.5) | 0.45 (0.39–0.51) | 697 (7.1) | 0.78 (0.57–1.07) |
| Blue collar | 3,678 (37.0) | 0.67 (0.59–0.76) | 1,771 (18.0) | 0.92 (0.76–1.11) |
| **Marriage status, n (%)** | | | | |
| Single | 52 (0.5) | 1.10 (0.57–2.11) | 106 (1.1) | 0.74 (0.31–1.79) |
| Married | 9,513 (95.7) | 1 | 7,660 (78.0) | 1 |
| Divorced/separated | 170 (1.7) | 1.15 (0.81–1.62) | 410 (4.2) | 1.37 (0.98–1.91) |
| Widowed | 209 (2.1) | 2.04 (1.57–2.65) | 1,650 (16.8) | 2.02 (1.71–2.37) |

Age is shown in mean value ± standard deviation (SD).

[a] Regular drinker: person drinking alcoholic beverages at least once a week.

[b] Exercise: exercise at least three times a week and more than 20 minutes at one time.

[c] Diabetes: plasma glucose ≥ 126mg/dL, previous history of diabetes

[d] Hypertension: systolic blood pressure ≥ 140mmHg or diastolic blood pressure ≥ 90mmHg at examination, previous history of hypertension, or current administration of antihypertensive (anti-HTN) medications.

[e] BP: Blood pressure.

[f] BMI: Body mass index.

[g] Prognostic Nutritional Index (PNI): calculated as 10 x serum albumin(g/dL) + 0.005 x total lymphocyte count (/mL), scored as 0(≥ 45) or 1(< 45).

[h] VO2max: maximum O2 uptake was measured by graded exercise test with bicycle ergometer, and it categorized as low (VO2max ≤ 21mL/kg/min for men / VO2max ≤ 10mL/kg/min for women), moderate (21mL/kg/min ≤ VO2max ≤ 27mL/kg/min for men / 10mL/kg/min ≤ VO2max ≤ 18mL/kg/min for women), high (VO2max ≥ 28mL/kg/min for men / VO2max ≥ 19mL/kg/min for women).

[i] BEPSI-K (Brief Encounter Psychosocial Instrument, Korean version), with results categorized as low (BEPSI-K < 1.8), moderate (1.8 ≤ BEPSI-K < 2.8), and high (BEPSI-K ≥ 2.8)

**Table 4. Adjusted hazard ratios (with 95% CIs) for all-cause and specific causes of mortality according to SRH by gender.**

| | All-cause mortality (HR, 95% CI) | Specific cause of mortality | | | |
| --- | --- | --- | --- | --- | --- |
| | | Cancer | Cardiovascular disease | Respiratory disease | Others |
| **Model 1[a]** | | | | | |
| **Male** | | | | | |
| Very good | 1 | 1 | 1 | 1 | 1 |
| Good | **1.26** (0.99–1.60) | **1.12** (0.80–1.56) | **0.93** (0.54–1.60) | **3.18** (0.74–13.57) | **1.65** (1.01–2.69) |
| Fair | **1.34** (1.06–1.69) | **1.11** (0.80–1.54) | **1.20** (0.72–2.01) | **4.59** (1.11–19.04) | **1.63** (1.01–2.63) |
| Poor | **2.13** (1.65–2.77) | **1.65** (1.14–2.40) | **2.25** (1.28–3.96) | **12.11** (2.83–51.77) | **2.13** (1.25–3.62) |
| **Female** | | | | | |
| Very good | 1 | 1 | 1 | 1 | 1 |
| Good | **0.80** (0.52–1.24) | **2.25** (0.89–5.69) | **0.41** (0.19–0.87) | **0.24** (0.06–1.01) | **0.73** (0.31–1.75) |
| Fair | **0.88** (0.59–1.31) | **2.19** (0.89–5.39) | **0.39** (0.20–0.75) | **0.19** (0.06–0.63) | **1.17** (0.53–2.56) |
| Poor | **1.07** (0.71–1.61) | **2.59** (1.04–6.46) | **0.40** (0.20–0.78) | **0.14** (0.04–0.52) | **1.76** (0.79–3.90) |
| **Model 2[b]** | | | | | |
| **Male** | | | | | |
| Very good | 1 | 1 | 1 | 1 | 1 |
| Good | **1.25** (0.98–1.60) | **1.11** (0.80–1.56) | **0.93** (0.54–1.61) | **3.16** (0.74–13.51) | **1.64** (1.00–2.67) |
| Fair | **1.32** (1.04–1.67) | **1.10** (0.79–1.52) | **1.18** (0.70–1.98) | **4.35** (1.05–18.10) | **1.60** (0.99–2.58) |
| Poor | **2.05** (1.58–2.66) | **1.62** (1.12–2.36) | **2.16** (1.22–3.83) | **10.68** (2.48–45.97) | **2.03** (1.19–3.46) |
| **Female** | | | | | |
| Very good | 1 | 1 | 1 | 1 | 1 |
| Good | **0.81** (0.53–1.25) | **2.27** (0.90–5.72) | **0.42** (0.20–0.90) | **0.23** (0.05–0.99) | **0.74** (0.31–1.78) |
| Fair | **0.89** (0.60–1.33) | **2.21** (0.90–5.44) | **0.40** (0.21–0.77) | **0.17** (0.05–0.57) | **1.21** (0.55–2.67) |
| Poor | **1.09** (0.72–1.65) | **2.68** (1.07–6.69) | **0.41** (0.21–0.81) | **0.12** (0.03–0.43) | **1.84** (0.83–4.09) |
| **Model 3[c]** | | | | | |
| **Male** | | | | | |
| Very good | 1 | 1 | 1 | 1 | 1 |
| Good | **1.22** (0.95–1.56) | **1.06** (0.64–1.60) | **0.88** (0.51–1.51) | **2.95** (0.69–12.64) | **1.69** (1.02–2.79) |
| Fair | **1.26** (1.00–1.60) | **1.02** (0.66–1.60) | **1.11** (0.66–1.87) | **4.24** (1.02–17.66) | **1.62** (0.99–2.65) |
| Poor | **1.97** (1.51–2.56) | **1.52** (0.93–2.50) | **2.11** (1.19–3.74) | **10.30** (2.39–44.44) | **1.98** (1.14–3.41) |
| **Female** | | | | | |
| Very good | 1 | 1 | 1 | 1 | 1 |
| Good | **0.83** (0.54–1.29) | **2.37** (0.93–6.01) | **0.47** (0.22–1.00) | **0.28** (0.06–1.25) | **0.72** (0.30–1.72) |
| Fair | **0.92** (0.62–1.38) | **2.22** (0.90–5.49) | **0.45** (0.23–0.87) | **0.19** (0.05–0.69) | **1.28** (0.58–2.81) |
| Poor | **1.14** (0.75–1.72) | **2.58** (1.03–6.48) | **0.49** (0.24–0.98) | **0.15** (0.04–0.57) | **1.98** (0.89–4.40) |

[a] Model 1: Adjusted for age, BMI, smoking, drinking, hypertension, diabetes, exercise, marriage, education, income, and jobclass

[b] Model 2: Adjusted for age, BMI, smoking, drinking, hypertension, diabetes, exercise, marriage, education, income, jobclass, PNI, VO2max, BEPSI-K, and Sleep time

[c] Model 3: Adjusted for age, BMI, smoking, drinking, hypertension, diabetes, exercise, marriage, education, income, jobclass, PNI, VO2max, BEPSI-K, Sleep time, cancer hx, total cholesterol, FBS, and GFR

Model 2, the variables included those in Model 1 plus PNI, VO2max, BEPSI-K, and sleep time, and in Model 3, the variables included those in Model 2 plus cancer history and laboratory data. In men, the aHR of all-cause mortality increased as SRH worsened after adjustment in all three models (aHR:1.22, 95% CI 0.95–1.56; aHR:1.26, 95% CI 1.00–1.60; aHR:1.97, 95% CI 1.51–2.56: good, fair, and poor SRH in Model 3, respectively), although the numerical values of total mortality risk were gradually attenuated from Model 1 to Model 3 (the aHR of poor SRH was 2.13 vs 2.05 vs 1.97 in Model 1, Model 2, and Model 3, respectively).

These results were similar in men for cancer mortality and respiratory disease mortality (1.65 vs 1.62 vs 1.52, aHRs of cancer mortality from Model 1 to Model 3; 12.11 vs 10.68 vs 10.30, aHRs of respiratory disease mortality from Model 1 to Model 3). The aHR of cardiovascular disease mortality for poor SRH was also significantly higher than that for very good SRH (aHR: 2.11, 95% CI 1.19–3.74 in Model 3). However, in women, the relationships between the risk of all-cause mortality and SRH were all insignificant (aHR:0.83, 95% CI 0.54–1.29; aHR:0.92, 95% CI 0.62–1.38; aHR:1.14, 95% CI 0.75–1.72, for good, fair, and poor SRH in Model 3, respectively). Interestingly, compared to very good SRH, the aHRs for poor SRH for cancer mortality in women were almost twice as high as in men (aHR of poor SRH in Model 3: 1.52 [95% CI 0.93–2.50] vs 2.58 [95% CI 1.03–6.48] for men and women, respectively). In men, the aHRs of cardiovascular and respiratory disease mortality for poor SRH were significantly higher than for very good SRH, whereas the aHRs of cardiovascular and respiratory diseases mortality for poor SRH in women were significantly lower than for very good SRH (aHR of poor SRH in Model 3: 0.49 [95% CI 0.24–0.98], 0.15 [95% CI 0.04–0.57], for cardiovascular diseases and respiratory diseases, respectively). The gender differences between SRH and all-cause and cause-specific mortality are shown in Fig 1.

## Discussion

In line with previous studies, in this retrospective cohort study we found a significant association between SRH and mortality. However, this association differed by gender and the specific cause of death. In terms of gender, men had a higher all-cause mortality rate as the evaluation of SRH worsened. On the other hand, women showed no statistically meaningful relations. From the perspective of specific causes of death, men and women showed large differences, especially in mortality from CVD and respiratory diseases. In men, as with all-cause mortality, the risks of mortality due to cancer, CVD, and respiratory diseases were higher for poor SRH than for very good SRH. Meanwhile, the risk of cancer mortality in women with poor SRH compared with those with very good SRH was almost twice as high as in men. However, the mortality risk from CVD and respiratory diseases in women with poor SRH was significantly lower than that in women with very good SRH (HR 0.49, 95% CI 0.24–0.98).

With regard to gender differences, there are inconsistencies across previous studies. Some studies have suggested that the effect of SRH on mortality is stronger for men but not for women. Spiers et al. reported that the predictive effect of SRH was stronger in men than women [20]. Grant and colleagues also presented evidence that the stable negative association of poor SRH and mortality on men disappeared in women over time [21]. On the other hand, several studies have suggested that the impact of SRH on mortality is stronger in women than in men. Onawola and colleagues showed that SRH had a relation with mortality only for women, not for men [8], and several other studies have suggested no gender differences in the SRH-mortality relationship [22–24].

In our study, gender differences in the effect of SRH on mortality are evident. First, the baseline characteristics by SRH in Table 1 show that the factors affecting SRH differ slightly by gender. Women are more likely than men to report their SRH as being worse. Most of the women were nonsmokers, and their smoking habit did not vary by SRH. However, more than half of the men had experience of smoking, and the worse the SRH, the more current smokers there were. Idler and Benyamini suggested that SRH is influenced by a healthy lifestyle, which affects health status through such factors as smoking or low drug compliance [6]. Mandernacka also suggested that healthy lifestyles are important factors in health assessments [25]. Although we performed the analysis after adjusting for the effect of smoking, the residual confounder of smoking might have influenced the gender differences in the effect of SRH on total

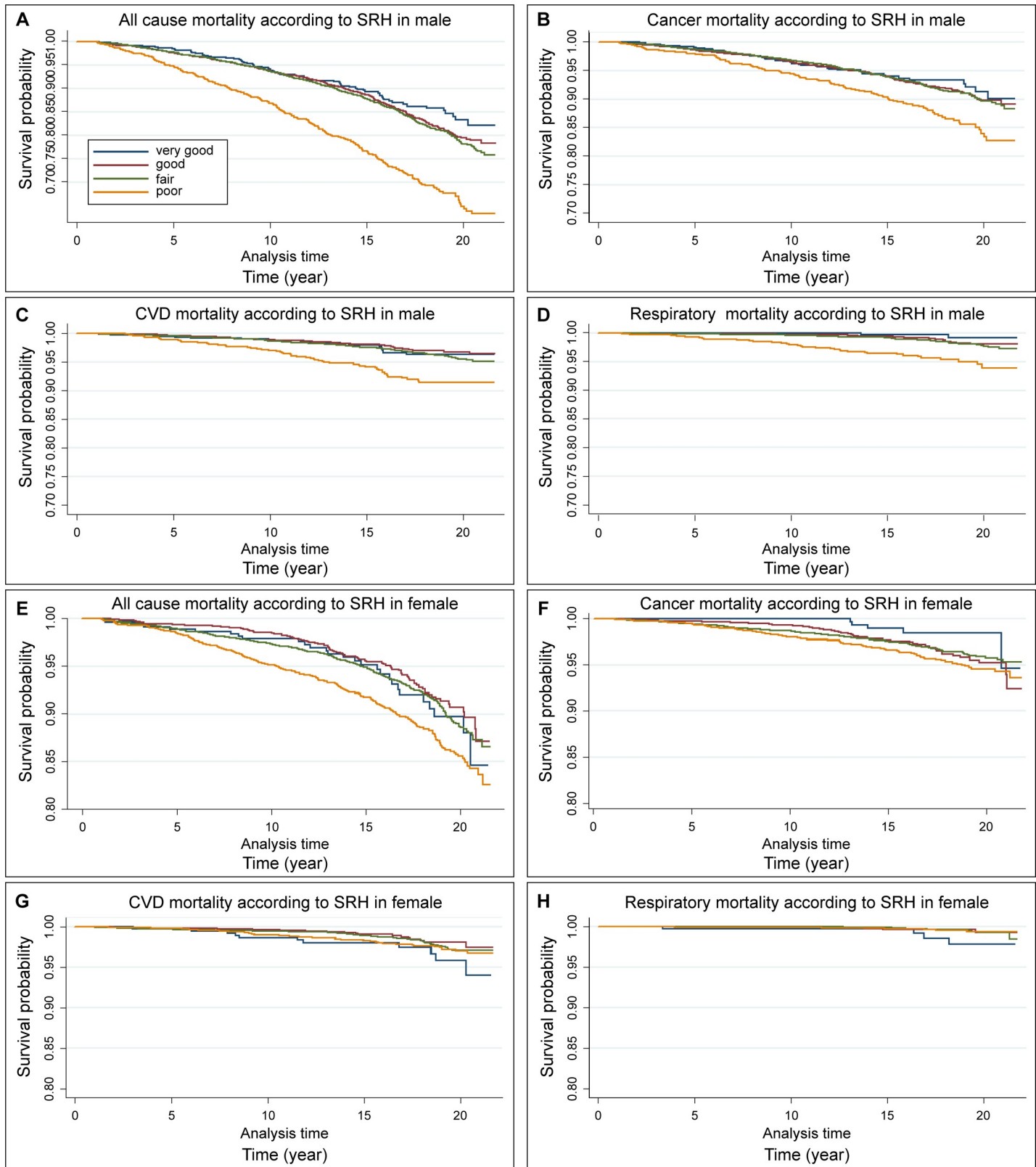

**Fig 1. Kaplan-Meier curve of all-cause and specific cause of mortality according to SRH by gender.** (A) All-cause of mortality of males (B) Cancer mortality of males (C) Cardiovascular disease mortality of males (D) Respiratory disease mortality of males (E) All cause of morality of females (F) Cancer mortality of females (G) Cardiovascular disease mortality of females (H) Respiratory disease mortality of females.

death. Thus, smoking status, which is more relevant to males than females, might have had a greater impact on males than females in assessing their health status, and a simple difference in these healthy lifestyles can cause gender-specific differences in SRH and its effect on mortality. Another difference was that in men there was no difference in the diagnosis of HTN by SRH, but in women, the worse the SRH, the more their history of being diagnosed as having hypertension. This could reflect a different attitude toward evaluating SRH by gender. Males tend to reflect mainly serious and life-threatening diseases, while females tended to reflect life-threatening as well as non-life-threatening diseases [21]. Females seemed more likely to include mild or chronic diseases in their general health assessment than males; therefore, the presence of higher blood pressure could have played an important role in assessing current health in females. One study showed a relation between SRH and hypertension in a Korean population, reporting that the relation of SRH and hypertension, which may be considered a typical chronic disease, was stronger in women than in men [26].

Second, the effect of SRH on all-cause mortality differed by gender. All-cause mortality was low in cases of good health behavior, better socio-epidemiological background, and healthier clinical data in both genders according to Table 3. For example, the lower the nutritional level, the higher the stress level, and the greater the history of diagnosis with hypertension, diabetes, and cancer, the higher the HRs in both genders. The subjects aged 65 years and over showed higher the HR than those aged between 50–64 years. However, only the effect of SRH on all-cause mortality showed a difference between males and females. The risk of all-cause mortality increased as SRH worsened in males, but there was no difference in the risk of all-cause mortality by SRH in females.

There are several explanatory theories for the gender differences in the relation of SRH and mortality, although the precise mechanism has not been elucidated. Wolinsky and Tierney proposed a "sponge" hypothesis to explain the relations of SRH and mortality in females [27], whereby women have a higher awareness of their physical symptoms and their reports of chronic disease and symptoms are fairly accurate. If so, SRH supplementation would not be necessary to predict mortality well in females, and the phenomenon thus entails weaker associations between SRH and mortality in females if health state is controlled. Another explanation is that different morbidity patterns among genders might be responsible. While chronic disease states or health and functional impairment occurred before death in females, more acute and severer illnesses were common in males [28]. Thus, although most of the males rate their SRH higher than females do during their lifetime, males experience a steeper mortality rate than females. Therefore, decline in SRH better predicts mortality for men than for women [29]. Similarly, as males have a shorter life expectancy than females, who live longer while enduring disability and ill health, if males recognize their health to be poor, they are more likely to be closer to death than women who believe their health is poor [30]. In addition, another possible mechanism is that females consider a wider, more inclusive range of health-related sources, even including family health status or socially desirable answers, when evaluating their state of health [27, 31], which could rather hinder the exact evaluation of their state of health and weaken its association with mortality.

Lastly, there was a difference in the effects of SRH on the specific cause of mortality by gender. Unlike males with consistently high mortality in cases of poor SRH, in females, CVD deaths were in fact lower for poor SRH, and cancer deaths based on SRH status were about twice as high as in males. This is the first study to show significantly higher CVD deaths in women with very good SRH than with poor SRH.

Several studies have shown that inflammatory cytokines affect subjective health determinations [32], and people who are depressed or exhausted show high levels of circulating and stimulated cytokines [33, 34]. Similarly, one study showed that both SRH and vital exhaustion were

positively correlated with the level of pro-inflammatory markers such as IL-6 and hs-CRP, which could lead to inflammation resulting in cancer or cardiovascular diseases [35]. We therefore assumed that females who consider themselves in poor health may have several chronic diseases, poor physical condition, and vital exhaustion, which could increase the risk of inflammation that might be associated with an increased incidence and mortality of cancers.

Unlike cancer mortality, however, the interpretation of CVD mortality in females may be slightly different. One study found that in women, the Duke Activity Status Index (DASI), which reflects one's fitness level, i.e., degree of functional impairment, attenuated the association of SRH and CVD events. However, there was a positive relation between SRH and CVD events when adjusted for demographic factors or coronary- and arterial-disease-related risk factors. This means that women's CVD events are affected not only by SRH and objective cardiovascular risk factors, but also by functional impairment levels [36]. Because of their cultural background, Korean women have to perform a variety of daily activities in occupational work as well as housework, whereas men usually have a certain occupational activity boundary. Also, Korean women who consider themselves in very good health may have a tendency to perform unreasonably excessive work and activities. Thus, mental and physical overwork of women who consider themselves healthy could lead to fatal fatigue and functional impairment. These can cause pro-thrombotic and inflammatory reactions, which can increase the risk of vascular diseases, possibly related to sudden death, such as CVD [37, 38]. However, these explanations are somewhat elusive, and further research is needed to clarify the relationship between SRH and specific causes of death in females. On the other hand, the gender-specific risk differences in respiratory disease mortality should be interpreted cautiously and subjected to further investigation because there were relatively few deaths from respiratory diseases in men.

Our research has several strengths. First, we sought to show the association of SRH and mortality by considering various confounders such as social-demographic factors (age, gender, marital status, education job class, income level), health-related behavioral factors (smoking status, alcohol consumption, physical activity, nighttime sleep duration), and even clinical factors (PNI, VO2max, BEPSI, BMI, results of laboratory tests). These increased the reliability of our results regarding SRH and mortality. Second, our study population was large enough for analyses of specific causes of mortality by gender according to SRH. In addition, we studied adults over 50 years of age, whose health assessment are highly correlated with follow-up health outcomes.

However, the present study has some limitations. Our study subjects were collected from single-centered hospitals comprising a single ethnic group, which limits generalizability. Also, SRH was measured only once, at the baseline. Therefore, we could not demonstrate an association between changes in SRH over time and the risk for mortality. Although we controlled various confounders in the statistical models, we could not have controlled all the effects of confounders.

Despite these limitations, we found that SRH was associated with all-cause mortality in men but not in women, and that a differential effect of SRH on specific causes of mortality was noted according to gender. Men with poor SRH consistently showed higher risks of all-cause mortality and death from cancer, CVD, and respiratory diseases than did those with very good SRH. However, women with poor SRH showed a higher risk of cancer death but a lower risk of CVD death than did those with very good SRH. Thus, it is appropriate for the clinician to be aware of this gender difference and take it into consideration in practice.

## Author Contributions

**Conceptualization:** Insun Ryou, Minseon Park.

**Data curation:** Insun Ryou, Minseon Park.

**Formal analysis:** Insun Ryou, Yujin Cho, Hyung-Jin Yoon.

**Supervision:** Minseon Park.

**Writing – original draft:** Insun Ryou.

**Writing – review & editing:** Insun Ryou, Yujin Cho, Hyung-Jin Yoon.

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
