## [Decision Letter · Decision Letter 0]

22 Aug 2019

PONE-D-19-17249

Gender differences in the effect of self-rated health (SRH) on all-cause mortality and specific causes of mortality among individuals aged 50 years and older

PLOS ONE

Dear Dr. Pf Park,

Thank you for submitting your manuscript to PLOS ONE. After careful consideration, we feel that it has merit but does not fully meet PLOS ONE’s publication criteria as it currently stands. Therefore, we invite you to submit a revised version of the manuscript that addresses the points raised during the review process.

Both the Reviewer and the Editor feel that the authors omitted to provide some essential information. The main points are detailed in the Reviewer's comments. Please respond to all in detail.

We would appreciate receiving your revised manuscript by Oct 06 2019 11:59PM. To enhance the reproducibility of your results, we recommend that if applicable you deposit your laboratory protocols in protocols.io, where a protocol can be assigned its own identifier (DOI) such that it can be cited independently in the future. For instructions see: http://journals.plos.org/plosone/s/submission-guidelines#loc-laboratory-protocols

We look forward to receiving your revised manuscript.

Kind regards,

Andreas Zirlik, MD

Academic Editor

PLOS ONE

Journal Requirements:

2. Please include a copy of the questionnaires used in the study, in both the original language and English, as Supporting Information, or include a citation if it has been published previously.

Reviewers' comments:

Reviewer's Responses to Questions

**Comments to the Author**

1. Is the manuscript technically sound, and do the data support the conclusions?

Reviewer #1: Partly

2. Has the statistical analysis been performed appropriately and rigorously? 

Reviewer #1: I Don't Know

3. Have the authors made all data underlying the findings in their manuscript fully available?

Reviewer #1: Yes

4. Is the manuscript presented in an intelligible fashion and written in standard English?

Reviewer #1: Yes

5. Review Comments to the Author

Reviewer #1: In the manuscript the authors nicely diplay their data gathered from a retrospective cohort study to expose and analyse the gender specific differences of self-rated health (SRH) and and the relationship between SRH and all-cause mortality as well as cause specific mortality. In their statistical analysis the authors showed that there is an association between SRH and all-cause mortality as well as cause specific mortality after adjusting for various confounders in men. In women however an association was only shown for cancer related mortality while there was no association between SRH and all cause mortality. CVD associated mortality and respiratory disease mortality even showed an inverse relationship to SRH. With these findings the authors conclude that SRH and its relationship on all-cause and specific mortality differs between genders and that clinicians should take this into account.

Overall I think that the presented study can satisfy the criteria for publication in PLOS ONE if some issues are addressed which I will specify below.

1) In Table 2 SRH it sticks out, that event rates for mortality of all cause, cancer and others are higher in the "very good" SRH-group than in the good and fair rated group. I think it would be worthwhile to analyse these incidence rates specific by gender to identify if the contraintuitive finding that better SRH is accompanied by higher event rates is mainly driven by females (as the lower event rates in CVD and respiratory disease might indicate and especially as gender differences are the main conclusion of the manuscript).

2) In the sections where the authors describe the statistical analysis it is written that all causes of mortality are adjusted for age, body mass index (BMI), smoking, drinking, diagnosis of hypertension or diabetes, PNI, VO2max, BEPSI-K sleep time, history of cancer, total cholesterol level, fasting blood glucose level and GFR. Occupational status, educational status and income level are not in this list but seem rather important confounders when evaluating SRH. In the result section in table 4 however income, education and job class seem to be included in the adjusted confounders. I would encourage the authors to either specify in the statistics section if the mentioned confounders are included in the analysis or to redo the analysis with the confounders included.

3) In line 211 it says: "During a median follow-up 15,4 years 2,263 of the 11.770 individuals (19,2%) had died". In the methods section however it is said that 19,770 individuals where included in the study (9944 males, 9826 females which is consistent in the other tables), so i dont get which cohort was analysed regarding the incidence rates in table 2.

4) In the discussion section in line 317 the authors speculate that the SRH of men might be more affected by consideration their smoking status. However it is also mentioned repeatedly that the opinion in the field is that men reflect mainly serious and life threatening-disease (e.g. line 322-324) which seems contradictory. I would encourage the authors to discuss this contradiction more detailed.

5) In lines 320 to 329 the authors discuss wether a different wheighting of hypertension might be partly accountable for the gender differences in SRH. It is proposed that women tend to rate their health worse when having hypertension. This however is kind of contraintuitive to the results displayed in table 4 where women with poorer self rated health exibit lower rates of CVD related death and hypertension being one major driver of these CVD-related deaths.

6) In line 362 the mutual influcence of SRH and inflammatory state is discussed and the authors propose a chain of causality where females who consider themselves in poor health may have several chronic diseases, poor physical condition, and vital exhaustion, which could increase the risk of inflammation that might be associated with an increased incidence and mortality of cancers. While this seems reasonable in general it is suprising that only cancer related mortality is increasing and not CVD-related mortality as a proinflammatory state is known to drive CVD disease as well. Maybe the authors can add this into their considerations.

7) In line 333 it is said: "For examples, the older the age [...] the higher the HRs in both genders." The authors should reframe the sentence in "people over 65 show higher HR ... " because in the data they only separated between people >65 and 54-65.

8) I would encourage the authors to explain in their introduction why they excluded people which died within a year of follow-up.

9) Maybe it would be possible to illustrate the main finding of the manuscript (like the different HRs for the different SRH and genders) by a more graphic illustration for a easer visualisation of the main massage.

10) In line 312-314 it is said: "Idler and Benyamini suggested [...]." but the citation is from Guimaraes et al.

11) In the references (line 482) it says number 22 is an invalid citation.

12) In line 116 and 216 there are missing full stops.

13) In line 117 the full stop after "week" is wrong.

14) In line 214 cross out "in that order".

15) Sometimes there are blanks befor the % sing, sometimes not (e.g. lines 213/214). I would encourage the authors to do anothers proofsreading before finally submitting the manuscript.

6. PLOS authors have the option to publish the peer review history of their article (what does this mean?). If published, this will include your full peer review and any attached files.

Reviewer #1: No

---

## [Author Response · Author response to Decision Letter 0]

3 Oct 2019

I have responded specifically to each suggestion below. 

1) In Table 2 SRH it sticks out, that event rates for mortality of all cause, cancer and others are higher in the "very good" SRH-group than in the good and fair rated group. I think it would be worthwhile to analyse these incidence rates specific by gender to identify if the contraintuitive finding that better SRH is accompanied by higher event rates is mainly driven by females (as the lower event rates in CVD and respiratory disease might indicate and especially as gender differences are the main conclusion of the manuscript).

 -> As suggested, we changed Table 2, to better show that SRH is accompanied by higher event rates mostly driven by females, especially in CVD mortality. 

Because of an error in the calculation of the incidence rate due to an error in person year, we recalculated incidence rate by event/100,000 PY 

2)In the sections where the authors describe the statistical analysis it is written that all causes of mortality are adjusted for age, body mass index (BMI), smoking, drinking, diagnosis of hypertension or diabetes, PNI, VO2max, BEPSI-K sleep time, history of cancer, total cholesterol level, fasting blood glucose level and GFR. Occupational status, educational status and income level are not in this list but seem rather important confounders when evaluating SRH. In the result section in table 4 however income, education and job class seem to be included in the adjusted confounders. I would encourage the authors to either specify in the statistics section if the mentioned confounders are included in the analysis or to redo the analysis with the confounders included.

-> The mentioned confounders, i.e., occupational status, educational status, and income level, were already included in the main analysis.

We did not mention the mentioned confounders in the previous submission. Thus, this time we added the above explanation. 

3) In line 211 it says: "During a median follow-up 15,4 years 2,263 of the 11.770 individuals (19,2%) had died". In the methods section however it is said that 19,770 individuals where included in the study (9944 males, 9826 females which is consistent in the other tables), so i dont get which cohort was analysed regarding the incidence rates in table 2.

-> There was a mistake in typing the numbers. The total study population was 19,770 and the death toll was 2,263. Table 2 was calculated using these numbers. 

4) In the discussion section in line 317 the authors speculate that the SRH of men might be more affected by consideration their smoking status. However it is also mentioned repeatedly that the opinion in the field is that men reflect mainly serious and life threatening-disease (e.g. line 322-324) which seems contradictory. I would encourage the authors to discuss this contradiction more detailed.

5) In lines 320 to 329 the authors discuss wether a different wheighting of hypertension might be partly accountable for the gender differences in SRH. It is proposed that women tend to rate their health worse when having hypertension. This however is kind of contraintuitive to the results displayed in table 4 where women with poorer self rated health exibit lower rates of CVD related death and hypertension being one major driver of these CVD-related deaths.

-> Ellen L. Idler (The Gerontologist, Volume 43, Issue 3, June 2003, Pages 372–375) quoted a report by Deeg and Kriegsman (The Gerontologist, Volume 43, Issue 3, June 2003, Pages 376–386) as explaining that men’s SRH tends to take into account lifestyle factors and mortality risks, while women’s SRH is likely to be associated with disabling health conditions. 

Thus, we concluded that health-related lifestyle behaviors could not be compared with the severity of the disease. What we're trying to say is, there are gender-based differences in the factors considered in assessing health conditions; i.e., males consider health-related behaviors such as smoking that are known to be risk factors for several disease, unlike female. 

Although there is inconsistency regarding current smoking and poor SRH in several studies, lifestyle factors like smoking seem to be related with poor SRH. One study showed that the odds of good SRH were significantly higher among non-smokers than smokers in Estonia. (European Journal of Public Health, Volume 28, Issue suppl_4, November 2018) Another study reported that smokers in Korea had good SRH (Journal of Korean Medical Science, Volume 30, Issue 9, September 2015); however, looking closely at the study group of the study, the majority of nonsmoking group were women, and the majority of smoking group were men. According to our results, there was a tendency for females to assess their health badly, and for males to assess their health favorably. Since the study population above was not divided by gender, it appears that much of the poor SRH in the nonsmoking group was mainly due to women’s tendency to rate their SRH as poor. Similarly, it appears that a large portion of good SRH in the smoking group was mainly due to men’s tendency to rate their SRH as good.

Also, apart from the lifestyle, females who perceive their health status as bad have chronic diseases such as hypertension, unlike males, who are affected mostly by a life-threatening disease. 

However, because the SRH of females is not associated with mortality in our results, although hypertension itself could be the cause of CVD mortality, CVD mortality in female might be due to another factor than simply having hypertension, and an explanation of other factors will be given in Comment 6 as a possible response. 

6) In line 362 the mutual influcence of SRH and inflammatory state is discussed and the authors propose a chain of causality where females who consider themselves in poor health may have several chronic diseases, poor physical condition, and vital exhaustion, which could increase the risk of inflammation that might be associated with an increased incidence and mortality of cancers. While this seems reasonable in general it is suprising that only cancer related mortality is increasing and not CVD-related mortality as a proinflammatory state is known to drive CVD disease as well. Maybe the authors can add this into their considerations.

-> According to one study (Psychosomatic Medicine, Volume 72, Issue 6, July 2010), in women, the degree of functional impairment attenuated the association of SRH and CVD events, showing that women’s CVD events are affected not only by SRH and objective cardiovascular risk factors, but also by functional impairment levels. Thus, females who regard themselves as having very good health status and who perform unreasonably excessive work and activities can cause themselves to suffer transient functional impairment, for example hemodynamic instability, which could be a reason for their higher risk of CVD mortality.

7) In line 333 it is said: "For examples, the older the age [...] the higher the HRs in both genders." The authors should reframe the sentence in "people over 65 show higher HR ... " because in the data they only separated between people >65 and 54-65.

-> As suggested, we have made the following changes: 

For example, the lower the nutritional level, the higher the stress level, and the greater the history of diagnosis with hypertension, diabetes, and cancer, the higher the HRs in both genders. The subjects aged 65 years and over showed higher the HR than those aged between 50–64 years

8) I would encourage the authors to explain in their introduction why they excluded people which died within a year of follow-up.

-> We wanted to identify only the impact of SRH on mortality. However, there could have been deaths of patients who had stage 0 cancer, asymptomatic heart disease, or un-diagnosed unknown diseases at screening. To eliminate deaths from these causes, we excluded deaths within one year after the screening.

Above contents are described in the Ascertainment of Mortality subsection in the Materials and Methods section. 

9) Maybe it would be possible to illustrate the main finding of the manuscript (like the different HRs for the different SRH and genders) by a more graphic illustration for a easer visualisation of the main massage.

-> As suggested, we made the figure 1 

10) In line 312-314 it is said: "Idler and Benyamini suggested [...]." but the citation is from Guimaraes et al.

-> There was a mistake in typing the references. We have corrected it.

11) In the references (line 482) it says number 22 is an invalid citation.

-> There was a mistake in typing the references. We have corrected it.

12) In line 116 and 216 there are missing full stops.

-> We have corrected it.

13) In line 117 the full stop after "week" is wrong.

-> We have corrected it.

14) In line 214 cross out "in that order".

-> We have corrected it.

15) Sometimes there are blanks befor the % sing, sometimes not (e.g. lines 213/214). I would encourage the authors to do anothers proofsreading before finally submitting the manuscript.

 -> We have corrected it.

---

## [Decision Letter · Decision Letter 1]

12 Nov 2019

Gender differences in the effect of self-rated health (SRH) on all-cause mortality and specific causes of mortality among individuals aged 50 years and older

PONE-D-19-17249R1

Dear Dr. Park,

We are pleased to inform you that your manuscript has been judged scientifically suitable for publication and will be formally accepted for publication once it complies with all outstanding technical requirements.

With kind regards,

Andreas Zirlik, MD

Academic Editor

PLOS ONE

Additional Editor Comments (optional):

Reviewers' comments:

Reviewer's Responses to Questions

**Comments to the Author**

1. If the authors have adequately addressed your comments raised in a previous round of review and you feel that this manuscript is now acceptable for publication, you may indicate that here to bypass the “Comments to the Author” section, enter your conflict of interest statement in the “Confidential to Editor” section, and submit your "Accept" recommendation.

Reviewer #1: All comments have been addressed

2. Is the manuscript technically sound, and do the data support the conclusions?

Reviewer #1: Yes

3. Has the statistical analysis been performed appropriately and rigorously? 

Reviewer #1: I Don't Know

4. Have the authors made all data underlying the findings in their manuscript fully available?

Reviewer #1: Yes

5. Is the manuscript presented in an intelligible fashion and written in standard English?

Reviewer #1: Yes

6. Review Comments to the Author

Reviewer #1: (No Response)

7. PLOS authors have the option to publish the peer review history of their article (what does this mean?). If published, this will include your full peer review and any attached files.

Reviewer #1: No

---

## [Editor Report · Acceptance letter]

22 Nov 2019

PONE-D-19-17249R1 

Gender differences in the effect of self-rated health (SRH) on all-cause mortality and specific causes of mortality among individuals aged 50 years and older 

Dear Dr. Park:

I am pleased to inform you that your manuscript has been deemed suitable for publication in PLOS ONE. Congratulations! Your manuscript is now with our production department. 

With kind regards,

on behalf of

Univ. Prof. Dr. Andreas Zirlik 

Academic Editor

PLOS ONE